# Generation and comparison of CRISPR-Cas9 and Cre-mediated genetically engineered mouse models of sarcoma

Jianguo Huang[1], Mark Chen[2,3], Melodi Javid Whitley[2,3], Hsuan-Cheng Kuo[2], Eric S. Xu[1], Andrea Walens[2], Yvonne M. Mowery[1], David Van Mater[4], William C. Eward[5], Diana M. Cardona[6], Lixia Luo[1], Yan Ma[1], Omar M. Lopez[2], Christopher E. Nelson[7,8], Jacqueline N. Robinson-Hamm[7,8], Anupama Reddy[8], Sandeep S. Dave[8,9], Charles A. Gersbach[7,8], Rebecca D. Dodd[1,†] & David G. Kirsch[1,2]

Genetically engineered mouse models that employ site-specific recombinase technology are important tools for cancer research but can be costly and time-consuming. The CRISPR-Cas9 system has been adapted to generate autochthonous tumours in mice, but how these tumours compare to tumours generated by conventional recombinase technology remains to be fully explored. Here we use CRISPR-Cas9 to generate multiple subtypes of primary sarcomas efficiently in wild type and genetically engineered mice. These data demonstrate that CRISPR-Cas9 can be used to generate multiple subtypes of soft tissue sarcomas in mice. Primary sarcomas generated with CRISPR-Cas9 and Cre recombinase technology had similar histology, growth kinetics, copy number variation and mutational load as assessed by whole exome sequencing. These results show that sarcomas generated with CRISPR-Cas9 technology are similar to sarcomas generated with conventional modelling techniques and suggest that CRISPR-Cas9 can be used to more rapidly generate genotypically and phenotypically similar cancers.

[1] Department of Radiation Oncology, Duke University Medical Center, Durham, North Carolina 27710, USA. [2] Department of Pharmacology and Cancer Biology, Duke University Medical Center, Durham, North Carolina 27710, USA. [3] Medical Scientist Training Program, Duke University Medical Center, Durham, North Carolina 27710, USA. [4] Division of Hematology-Oncology, Department of Pediatrics, Duke University Medical Center, Durham, North Carolina 27710, USA. [5] Department of Orthopedic Surgery, Duke University, Durham, North Carolina 27710, USA. [6] Department of Pathology, Duke University, Durham, North Carolina 27710, USA. [7] Department of Biomedical Engineering, Duke University, Durham, North Carolina 27708, USA. [8] Duke Center for Genomic and Computational Biology, Duke University, Durham, North Carolina 27708, USA. [9] Department of Medicine, Duke University Medical Center, Durham, North Carolina 27710, USA. † Present address: Department of Internal Medicine, University of Iowa, Iowa City, Iowa 52242, USA. Correspondence and requests for materials should be addressed to D.G.K. (email: david.kirsch@duke.edu).

Genetically engineered mouse models (GEMMs) are important tools for studying cancer *in vivo*[1,2]. First-generation GEMMs relied on transgenic germline manipulation to express oncogenes. Subsequent GEMMs applied homologous recombination by targeting tumour suppressor genes in embryonic stem (ES) cells by introducing specific mutations into a defined genetic locus. When this approach was modified to permit regulation by site-specific recombinases (SSR), such as Cre recombinase, investigators gained spatial and temporal control over the development of cancer in mice[3]. We and many other laboratories have successfully used this approach to gain important insights into cancer and to study the response of primary tumours to cancer therapy[4,5]. However, the process of generating GEMMs is costly and time consuming, as it includes targeting ES cells, breeding mice, and genotyping the progeny. These drawbacks are compounded when more complex models are desired that involve conditional, inducible, or multiple alleles.

The discovery and adaptation of the prokaryotic clustered regularly interspaced short palindromic repeats (CRISPR) system for genome editing in eukaryotic cells and organisms, including mice, has expanded the *in vivo* disease modelling armamentarium[6–12]. In particular, CRISPR-Cas9 genome engineering has been utilized to create mouse models of cancers of the pancreas, liver and lung, among other organs[10,13–19]. Developing mouse models of soft tissue sarcoma, however, has remained challenging. First, soft tissue sarcomas arise from connective tissue and encompass a broad range of subtypes. For many sarcoma subtypes, the progenitor cells remain unknown and targeting the correct progenitor has been a barrier to modelling distinct sarcoma histologies in mice[20–22]. Second, soft tissue sarcomas are genetically diverse. Some sarcomas harbour simple translocations such as the t(11;22)(q24;q12) translocation commonly found in Ewing sarcoma, whereas others have complex genetics such as in malignant peripheral nerve sheath tumour (MPNST) and undifferentiated pleomorphic sarcoma (UPS)[23,24]. Although spatially and temporally restricted mouse models of UPS and MPNST have been generated using Cre-loxP technology[3,25], it has been challenging to generate robust preclinical models for some sarcomas such as Ewing sarcoma[23]. As sarcomas are a rare and under-studied group of tumours, there is a need for *in vivo* models that accurately recapitulate this spectrum of cancers. Here we apply the CRISPR-Cas9 system to generate autochthonous UPS in a GEMM and MPNST in wild-type 129/SvJ mice.

The use of CRISPR-Cas9 to generate models of cancer has increased due to its utility in rapidly editing somatic cells. As CRISPR-Cas9 mouse models become more prevalent in pre-clinical research, it is important to carefully evaluate how these models compare with conventional models such as those generated using SSR technology. For example, traditional cancer models generated using Cre recombinase can provide temporal and spatial control for tumour development, but require a relatively long and costly process to genetically engineer the germline DNA. With CRISPR-Cas9 technology, somatic mutations can be quickly generated to initiate cancers with spatial and temporal control. Yet, as Cas9 is an efficient endonuclease, unintended off-target genomic changes may have an impact on *in vivo* phenotypes. To our knowledge, cancer mouse models generated by CRISPR-Cas9 and SSRs have not been systematically compared. Here we compare two mouse models of UPS generated with either Cre-loxP or CRISPR-Cas9 technology and find similar histology, growth kinetics and mutational profiles between these models.

## Results

### A combined Cre-loxP-CRISPR-Cas9 system for genome editing.
We have previously shown that intramuscular delivery of an adenovirus expressing Cre recombinase into mice with conditional mutations in *Kras* and *Trp53* results in the formation of soft tissue sarcomas at the site of injection[3]. To recapitulate this model using a combined Cre-loxP and CRISPR-Cas9 system, we crossed $Kras^{LSL-G12D/+}$ (K) mice with $Rosa26^{LSL-Cas9-EGFP/+}$ (C) mice to generate $Kras^{LSL-G12D/+}$; $Rosa26^{LSL-Cas9-EGFP/+}$ (KC) mice. KC mice conditionally express oncogenic *Kras* and *Cas9* in the presence of Cre recombinase through removal of upstream floxed STOP cassettes (LSL)[10,26]. We constructed the pX333-sgTrp53-Cre (pX333-P-Cre) plasmid to deliver both Cre recombinase and a single-guide RNA (sgRNA) targeting the mouse *Trp53* exon 7 (Fig. 1a and Supplementary Table 1). To assess the applicability of a combined Cre-loxP and CRISPR-Cas9 system for oncogenic transformation *in vitro*, mouse embryonic fibroblasts (MEFs) from K mice, C mice and KC mice were transiently transfected with the pX333-P-Cre plasmid and subjected to DNA analysis, soft agar assay and an *in vivo* allograft experiment (Fig. 1b). PCR analysis of genomic DNA from transfected K MEFs revealed activation of $Kras^{G12D}$ alone, from transfected C MEFs revealed activation of *Cas9* alone and from transfected KC MEFs revealed activation of both $Kras^{G12D}$ and *Cas9* (Fig. 1c). In addition, insertions or deletions (indels) in *Trp53* were detected in transfected C MEFs and KC MEFs by Surveyor nuclease assay, but not in K MEFs as expected (Fig. 1c). Activation of oncogenic *Kras* and deletion of *Trp53* were both required for anchorage-independent growth (Fig. 1d) and to form tumours resembling sarcoma following intramuscular injection into nude mice (Fig. 1e). Thus, these results show that a combined Cre-loxP and CRISPR-Cas9 system, which activates $Kras^{G12D}$ and mutates *Trp53 in vitro*, is sufficient for transformation.

### Generation of primary soft tissue sarcomas using CRISPR-Cas9.
Having achieved transformation by transient transfection of the pX333-P-Cre plasmid in KC MEFs *in vitro*, we next explored the possibility of generating primary sarcomas in KC mice by intramuscular injection of Adeno-sgTrp53-Cre (Ad-P-Cre), an adenovirus expressing sgTrp53 and Cre recombinase (Fig. 2a). After transduction of Ad-P-Cre adenovirus into KC MEFs, $Kras^{G12D}$ and *Cas9* were activated and *Trp53* indels were detected by Surveyor nuclease assay (Supplementary Fig. 1A). We then injected K mice ($n = 10$), C mice ($n = 10$), and KC mice ($n = 20$) intramuscularly with Ad-P-Cre adenovirus. Following Ad-P-Cre injection, we observed tumours in 100% of KC mice with tumour onset as early as 7.3 weeks (Fig. 2b). Of note, the time frame of Ad-P-Cre adenovirus-initiated tumour development (median 9.6 weeks) was similar to that of tumours generated in $Kras^{LSL-G12D/+}$; $Trp53^{Flox/Flox}$ (KP) mice by intramuscular injection of Ad-Cre (median 11.3 weeks)[3]. The virus did not generate sarcomas in mice harbouring only $Kras^{LSL-G12D/+}$ or only the $Rosa26^{LSL-Cas9-EGFP/+}$ allele (Fig. 2b). PCR confirmed $Kras^{G12D}$ activation and Surveyor assay showed evidence of *Trp53* indels in three cell lines derived from Ad-P-Cre-initiated tumours (Fig. 2c). *Trp53* complete knockout was also confirmed by western blotting, which showed no Trp53 expression induced by doxorubicin (Supplementary Fig. 1B)[27]. Interestingly, wild-type *Kras*, as well as activated *Cas9*, were undetectable in these tumour cells (Fig. 2c, Supplementary Fig. 1C). Loss of both the wild-type *Kras* allele and the adjacent *Cas9* allele may be selected for during sarcomagenesis because the wild-type *Kras* allele can act as a tumour suppressor in the presence of an oncogenic *Kras* that drives transformation and tumorigenesis[28–30]. Similar to sarcomas generated in Ad-Cre injected KP mice[31], immunohistochemical analysis of 12 Ad-P-Cre-induced tumours demonstrated that the majority of tumours ($n = 10$) are UPS without myogenic staining, whereas a minority ($n = 2$) were myogenic UPS (Fig. 2d,e).

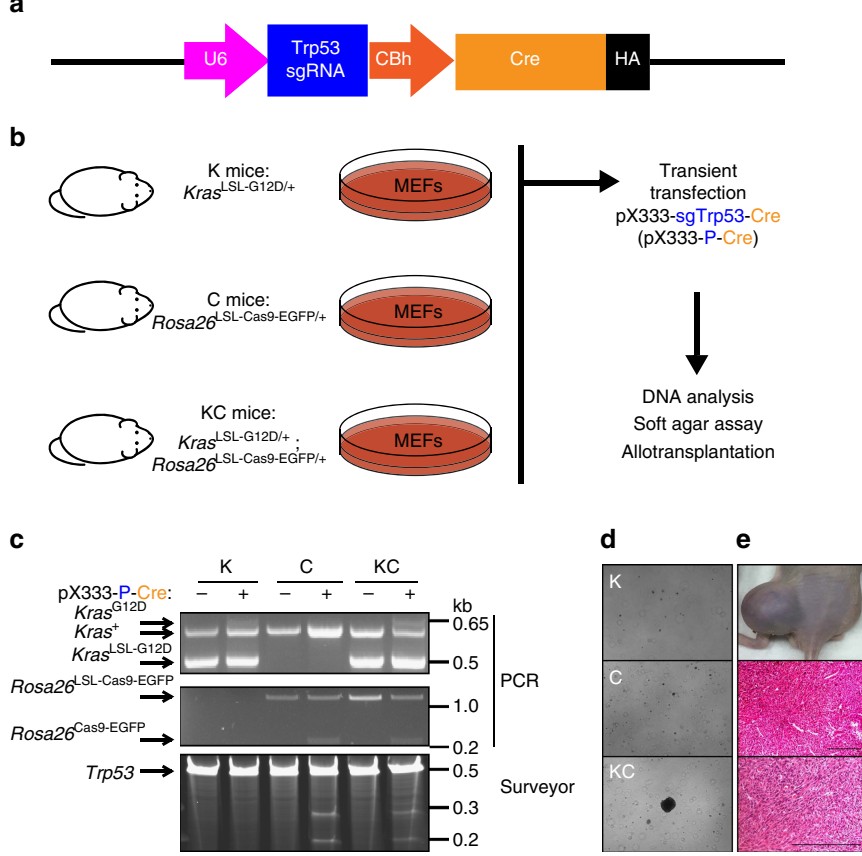

**Figure 1 | *In vitro* and *in vivo* validation of pX333-sgTrp53-Cre.** (**a**) The pX333 plasmid was modified to express Cre. A sgRNA to target *Trp53* was cloned into the plasmid following a U6 promoter. (**b**) MEFs from *Kras*[LSL-G12D/+] (K), *Rosa26*[LSL-Cas9-EGFP/+] (C) or *Kras*[LSL-G12D/+]; *Rosa26*[LSL-Cas9-EGFP/+] (KC) mice were harvested and cultured. MEFs were transiently transfected with pX333-sgTrp53-Cre and subsequently DNA analysis, soft agar studies and an allograft study were performed. (**c**) PCR shows that transient transfection of pX333-sgTrp53-Cre in MEFs resulted in recombination of *Kras*[LSL-G12D/+] and *Rosa26*[LSL-Cas9-EGFP/+] (top two panels). Surveyor assay (bottom panel) shows that pX333-sgTrp53-Cre efficiently generates indels in the *Trp53* gene. (**d**) MEFs transfected with pX333-sgTrp53-Cre were subjected to a soft agar assay. When KC MEFs were transfected with the plasmid, deletion of *Trp53* and activation of the conditional *Kras* mutation resulted in anchorage-independent growth. The result is representative of at least two different experiments. Furthermore, when these transformed cells were injected into nude mice (*n* = 4) (**e**), they all formed tumours that showed a spindle cell morphology. Scale bars, 100 μm.

**Assessment of clonality and mutations**. Non-homologous end joining repair following Cas9-induced double-strand breaks results in multiple indels in the targeted *Trp53* exon 7 (ref. 8). We assessed the clonality of sarcomas generated by Ad-P-Cre injection into KC mice by quantifying the number of unique mutations in exon 7 of *Trp53* in each tumour. To define the indels in *Trp53* in primary sarcomas, we performed Sanger sequencing of cell lines derived from each independent primary sarcoma (*n* = 4). Sanger sequencing demonstrated that no wild-type *Trp53* alleles were retained and indicated that three cell lines had at most two unique alterations for the two *Trp53* alleles (Fig. 3a), suggesting that these sarcoma cell lines arose from a single clone that underwent genome editing at both *Trp53* alleles. Interestingly, four distinct indels in the *Trp53* site were detected in one cell line (2995), suggesting that this sarcoma cell line either contained at least two clones or arose because of tetraploidization before mutagenesis, which is a frequent event in tumorigenesis[32–34]. This high frequency of biallelic modification could reflect high efficiency of on-target CRISPR-Cas9 gene modification, such that if one allele is modified it is highly likely for the other allele to be modified as well. Alternatively, cells with both *Trp53* alleles altered may be more likely or faster to contribute to sarcomagenesis. We investigated the first possibility by assessing the efficiency of CRISPR-Cas9 editing of a site on

chromosome 17 that is a perfect match for the sgRNA that we used to target Trp53 and that should not contribute to tumorigenesis. This site on chromosome 17 is an intergenic region located more than 125 kb away from the nearest protein coding gene (Supplementary Fig. 2A). Surveyor assay of the on-target chromosome 17 site showed evidence of indels in the Cas9 recognition site (Supplementary Fig. 2B). To define the indels at this site in primary sarcomas, we performed Sanger sequencing of the same four independent sarcoma cell lines (*n* = 4) from which we analysed *Trp53* modification. Sanger sequencing demonstrated that no wild-type sequence at the on-target chromosome 17 site was retained and indicated that each cell line had at most two unique alterations for this on-target chromosome 17 site (Fig. 3c). These data demonstrate that the high efficiency of genome editing by sgTrp53 is not restricted to genes required for transformation, suggesting that the CRISPR-Cas9 system could be utilized to target other genes to test their ability to modify sarcoma development and other phenotypes. As clonal analysis of derivative cell lines may introduce an additional level of *ex vivo* clonal selection, we also performed deep sequencing of *Trp53* and the on-target chromosome 17 sites in primary sarcomas (*n* = 5). Several deletions in the *Trp53* and on-target chromosome 17 sites identified by Sanger sequencing of the cell lines were too large to be detected by deep sequencing.

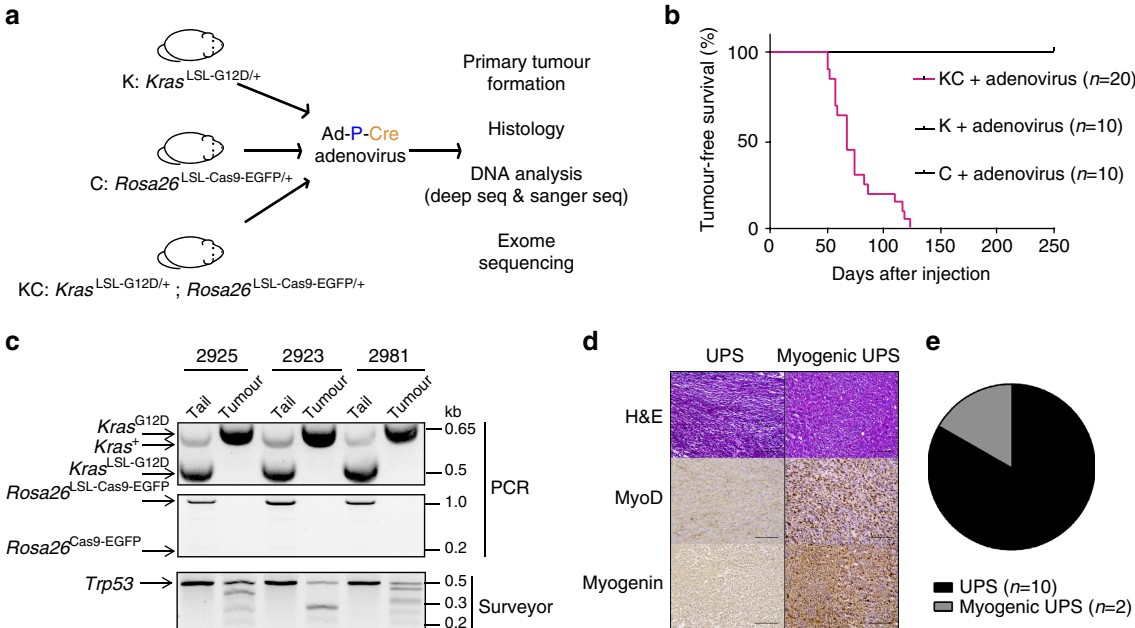

**Figure 2 | Conditional expression of Cas9 to generate tumours in a primary mouse model of sarcoma.** (**a**) *Kras*[LSL-G12D/+] (K), *Rosa26*[LSL-Cas9-EGFP/+] (C), or *Kras*[LSL-G12D/+]; *Rosa26*[LSL-Cas9-EGFP/+] (KC) mice received intramuscular injections of an adenovirus carrying the plasmid. (**b**) Adenoviral delivery of sgTrp3-Cre resulted in efficient primary sarcoma formation by activating conditional expression of the mutant *Kras* and *Cas9* alleles and delivering a sgRNA to target *Trp53*. (**c**) PCR analysis and Surveyor assay showing recombination of *Kras*[LSL-G12D] and *Rosa26*[LSL-Cas9-EGFP], as well as indels within the *Trp53* gene, respectively. (**d**) Twelve of the primary tumours generated via adenoviral plasmid delivery were subject to histopathological analysis with staining for myogenic markers MyoD and myogenin. (**e**) The majority of tumours resembled UPS, whereas a minority were consistent with a myogenic UPS phenotype. Scale bars, 500 μm.

This limitation of our deep sequencing was most likely to be due to primer design, which failed to amplify the targeted loci in the presence of a large indel that may have eliminated a complementary primer sequence required for PCR amplification. We still only identified a maximum of two distinct indels, defined as >1% of total reads by deep sequencing, in *Trp53* (Fig. 3b and Supplementary Table 2). Deep sequencing of the on-target chromosome 17 site also showed a maximum of two distinct indels in four out of five primary sarcomas (Fig. 3d and Supplementary Table 2). However, three distinct indels in the on-target chromosome 17 site were detected in one tumour (2995). This finding further indicated that this sarcoma contained either multiple clones or underwent tetraploidization before mutagenesis.

Although Cas9 precisely cleaved the targeted *Trp53* region and on-target chromosome 17 site with perfect complementarity to the sgRNA to produce site-specific indels in each tumour, several studies have shown that Cas9 activity may also result in off-target mutations in other regions of the genome[35,36]. We performed targeted deep sequencing to profile the top ten predicted off-target sites of the *Trp53* sgRNA[37]. No significant off-target indels were observed in five Ad-P-Cre-induced sarcomas when compared to normal muscle from the same mouse as a control (Supplementary Table 3). Only a single tumour contained an edited exonic off-target site (*Celsr1*). In addition, the Surveyor assay identified DNA mismatches in the tumour DNA that further Sanger sequencing determined to be SNPs rather than mutations (Supplementary Fig. 3). We analysed the remaining nine non-exonic off-target sites by Surveyor assay and Sanger sequencing (Supplementary Fig. 4). DNA mismatches were only detected in one off-target site by Surveyor assay, which were also confirmed as SNPs in both the tumour and normal DNA by Sanger sequencing. Taken together, these results indicate that Cas9 specifically cleaves the targeted region to produce site-

specific indels in each tumour without causing detectable off-target mutations in other regions of the genome.

**Primary sarcoma generation via *in vivo* electroporation.** Virus-mediated delivery was very efficient for tumour generation and matched the kinetics of KP tumours generated using Ad-Cre only (Fig. 2b). To further simplify this approach, we injected the pX333-P-Cre plasmid directly into muscle and observed tumours in 4 out of 15 KC mice with tumours forming as early as 13.1 weeks (Fig. 4a). The time frame for tumour development via naked plasmid injection was both delayed and less efficient compared to Ad-P-Cre virus delivery. However, electroporation of plasmid efficiently generated tumours (80% penetrance after median 10.8 weeks, Fig. 4) with similar kinetics and histology as delivery of Ad-P-Cre (median 9.6 weeks, Fig. 2b,d). *In vivo* electroporation (IVE) of naked plasmid pcDNA3-EGFP was ~38 times more efficient for enhanced green fluorescent protein (EGFP) delivery to muscle fibres compared with adenovirus delivery of EGFP, but this increased efficiency did not affect the penetrance or the kinetics of sarcoma formation (Supplementary Fig. 5). To further optimize this system, we generated *Kras*[LSL-G12D/+]; *Rosa26*[loxP-Cas9-EGFP/+] mice (K-loxP-C) that express Cas9 endogenously. After electroporation with pX333-P-Cre plasmid, the K-loxP-C mice developed sarcomas with 100% penetrance (Fig. 4). The time to tumour onset was similar to Ad-P-Cre tumours in KC mice, but occurred within a narrower window of time ranging from 7.0 to 10.7 weeks (median 7.4 weeks, Fig. 4).

**Primary MPNST generation in wild-type mice with CRISPR-Cas9.** We next investigated whether other mouse models of sarcoma could be generated using CRISPR-Cas9 technology. MPNSTs are aggressive sarcomas that arise from cells of the peripheral nerve

**Figure 3 | Sequencing of primary sarcomas generated using a combined CRISPR-Cas9-Cre-loxP system.** (**a**) Sanger sequencing of the *Trp53* gene in tumours generated in KC mice after adenoviral delivery of pX333-sgTrp53-Cre. Most of the sequenced alleles demonstrated deletions but there was a single nucleotide insertion observed in mouse 2981- allele 1. (**b**) Targeted deep sequencing of the *Trp53* gene in tumours generated in KC mice. (Blue = sgTrp53; green = PAM; red = insertion/deletion). (**c**) Sanger sequencing of the on-target chromosome 17 site in tumours generated in KC mice after adenoviral delivery of pX333-sgTrp53-Cre. Most of the sequenced alleles demonstrated deletions but there was a single nucleotide insertion observed in mouse 2981- allele 1 and two nucleotides insertion observed in mouse 2995 allele 2. (**d**) Targeted deep sequencing of the on-target chromosome 17 site in tumours generated in KC mice. (Blue = sgTrp53; green = PAM; red = insertion/deletion).

sheath. Loss-of-function mutations in the Ras regulatory gene *neurofibromin 1* (*Nf1*) and the tumour suppressor *Cdkn2a* (also known as *p16/p19* or *Ink4a/Arf*) are commonly found in MPNSTs[38]. We have previously described a Cre-loxP-driven model of MPNST generated in *Nf1*<sup>Flox/Flox</sup>; *Ink4a/Arf* <sup>Flox/Flox</sup> mice through Ad-Cre injection into the sciatic nerve[25]. In this model, the median time to tumour onset was 17.6 weeks (ranging from 8.9 weeks to 38.9 weeks). Mice with mutations in *Nf1* and *Trp53* in *cis* on chromosome 11 (*Nf1* $^{+/-}$; *p53* $^{+/-}$) also develop MPNSTs through spontaneous loss of heterozygosity of the wild-type *Nf1* and *Trp53* alleles[39]. Therefore, we used sgTrp53 in combination with a sgRNA targeting exon 7 of *Nf1* (sgNf1) (Supplementary Table 1), to determine whether the CRISPR-Cas9 system could generate MPNSTs in wild-type 129/SvJ mice. We constructed an adenoviral vector expressing sgNf1, sgTrp53

and Cas9 (Ad-NP-Cas9), and packaged the construct into adenoviral particles (Fig. 5a). *Nf1* and *Trp53* gene editing was achieved rapidly and efficiently upon transduction of Ad-NP-Cas9 adenovirus into NIH-3T3 mouse fibroblasts (Supplementary Fig. 6A). Infection of wild-type MEFs with Ad-NP-Cas9 adenovirus was sufficient for transformation, as determined by colony formation in soft agar (Supplementary Fig. 6B) and tumour formation in nude mice allografts (Supplementary Fig. 6C). To determine whether MPNSTs could be generated in wild-type mice using CRISPR-Cas9, Ad-NP-Cas9 adenovirus was injected into the sciatic nerves of wild-type 129/SvJ mice (*n* = 12). Mice were followed for 150 days after injection and we detected tumours in 10 of 12 mice (median time to tumour 14.4 weeks, range 10.3–36.6 weeks) (Fig. 5b). At 23 and 32.1 weeks, the other 2 mice were found dead with no tumours detected. *Nf1* and *Trp53*

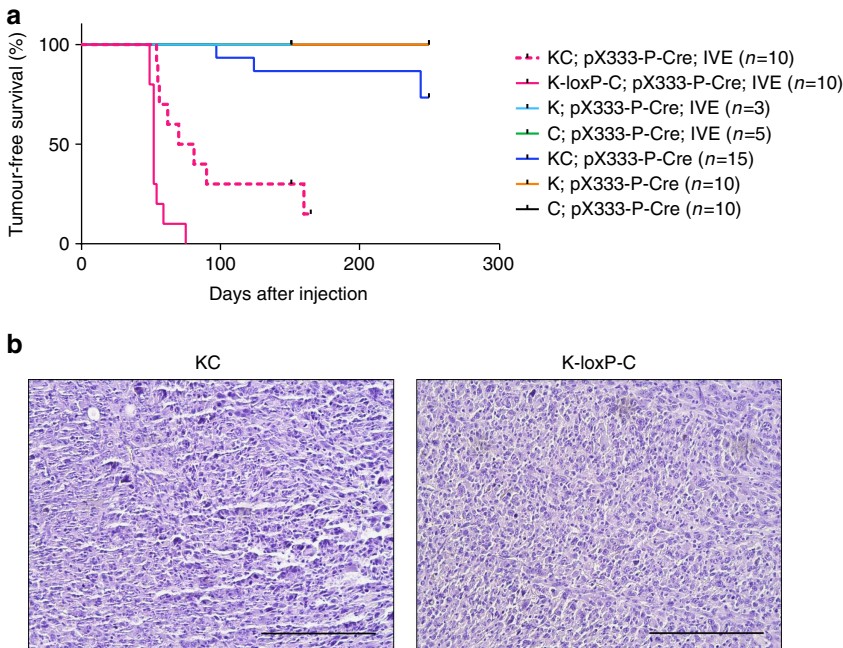

**Figure 4 | Generation of primary sarcomas using *in vivo* electroporation.** (**a**) K, C and KC mice received intramuscular injections of the naked pX333-sgTrp53-Cre and K, C, KC and K-loxP-C mice received intramuscular injections of the naked pX333-sgTrp53-Cre with IVE. (**b**) Ten of the primary tumours generated via IVE were subject to histopathological analysis with haematoxylin and eosin staining. Scale bars, 200 μm.

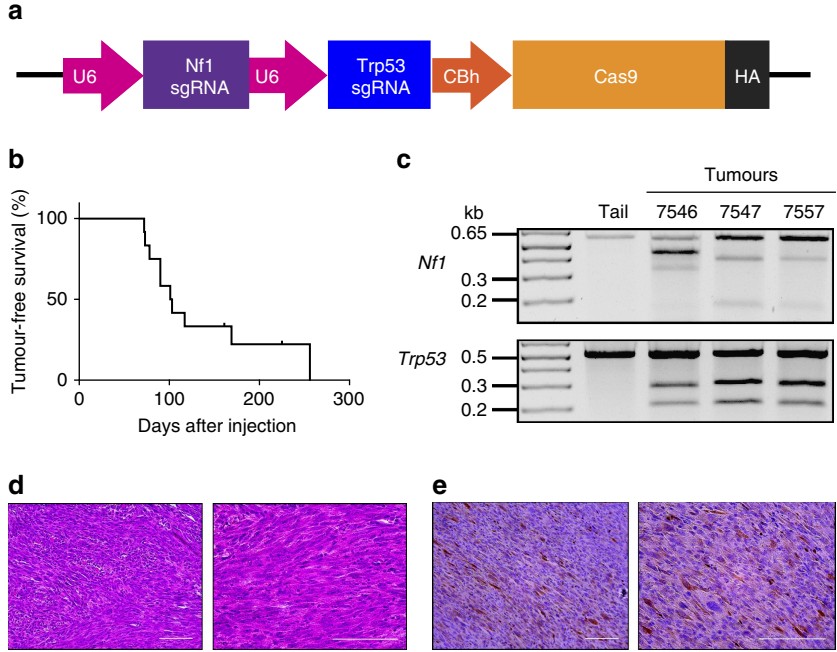

**Figure 5 | CRISPR-Cas9-mediated generation of primary MPNSTs.** (**a**) The px333 plasmid was modified by cloning a sgRNA targeting *Nf1* into the plasmid following the first U6 promoter and a sgRNA targeting *Trp53* following the second U6 promoter. (**b**) Wild-type 129/SvJ mice received sciatic nerve injections of an adenovirus carrying CRISPR components, and the mice were monitored for tumour formation. The formation of tumours was observed at the site of injection in these wild-type mice and (**c**) Surveyor analysis revealed indels in *Nf1* and *Trp53*. (**d**) Haematoxylin and eosin histology and (**e**) positive S100 staining were consistent with MPNST. Scale bars, 100 μm.

indels were detected by Surveyor assay in three of three cell lines derived from these tumours (Fig. 5c), and *Nf1* and *Trp53* knockout was confirmed by western blotting (Supplementary Fig. 6D). Furthermore, we performed histological analysis on five tumours revealing that each was an MPNST, with one tumour also containing myogenic features (Fig. 5d,e). Taken together, these data show that MPNSTs can be generated in

wild-type mice by applying CRISPR-Cas9 technology to target both *Nf1* and *Trp53* alleles.

**Comparison of CRISPR-Cas9 and Cre-mediated primary sarcomas.** The primary sarcomas initiated by CRISPR-Cas9-Cre-loxP and Cre-loxP alone resembled each other on haematoxylin and eosin-stained sections (Fig. 6a) and had similar growth kinetics as

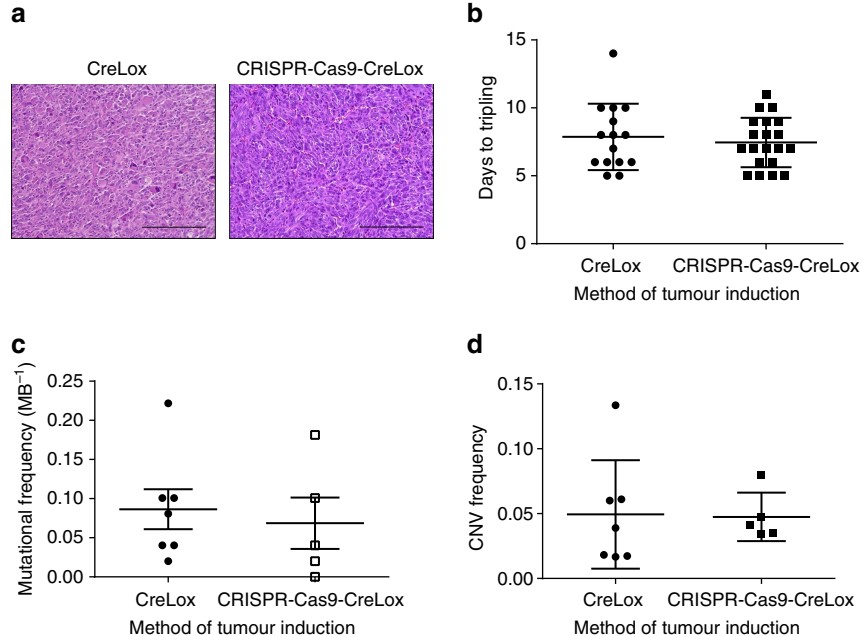

**Figure 6 | Whole exome sequencing of primary sarcomas generated using Cre-loxP or a combined CRISPR-Cas9-Cre-loxP system.** (**a**) Ten of the primary tumours generated via Cre-loxP or a combined CRISPR-Cas9-Cre-loxP system were subject to histopathological analysis with haematoxylin and eosin staining. (**b**) Comparison of the days to tripling of primary sarcomas generated with Cre-loxP ($n = 15$) and the combined CRISPR-Cas9-Cre-loxP ($n = 20$) system. Scale bars, 200 μm. The non-synonymous mutational load (**c**) and CNVs (**d**) of primary sarcomas generated with Cre-loxP ($n = 7$) and the combined CRISPR-Cas9-Cre-loxP ($n = 5$) system was determined using whole exome sequencing of tumour DNA (Error bars = s.e.m.).

measured by time to tumour tripling (median 8 days, Fig. 6b). To profile the mutational load and copy number variation (CNV) using an unbiased approach, we performed whole exome sequencing of primary mouse sarcomas (Fig. 6c,d). We compared the mutational load, which we defined as the number of non-synonymous somatic mutations per megabase (MB) of DNA, from tumours generated in KC mice following Ad-P-Cre injection ($n = 5$) and tumours generated in KP mice following Ad-Cre injection ($n = 7$). We also compared the amplifications and deletions to assess global CNV in these tumours. Overall, there were no significant differences in mutational load and CNV between sarcomas generated with or without CRISPR-Cas9 technology (Fig. 6c,d). Therefore, these results demonstrate that a combined Cre-loxP and CRISPR-Cas9 system that activates $Kras^{G12D}$ and deletes $Trp53$ in KC mice is sufficient to result in primary soft tissue sarcomas with mutational load and CNV similar to an established Cre-loxP tumour model.

## Discussion

Here we report, to the best of our knowledge, the first application of CRISPR-Cas9-mediated mutagenesis to generate autochthonous mouse models of sarcoma in GEMMs and wild-type mice. We generated primary UPS tumours *in vivo* using either Cre-loxP only or a combined Cre-loxP and CRISPR-Cas9 system. This hybrid system can be applied to existing mouse models of sarcoma for functional studies of genes mutated in human soft tissue sarcomas. Importantly, we compared the histopathology, tumour growth kinetics and mutational profiles of the tumours generated using these two systems, to determine that CRISPR-Cas9 generates similar mouse models of sarcoma compared to traditional SSR technology. In addition, we generated MPNSTs in wild-type mice with Cas9 and sgRNAs targeting *Nf1* and *Trp53*. These models decrease the expense and time to generate primary sarcomas, which should expand the opportunity to employ GEMMs by more investigators in cancer research.

Primary soft tissue sarcomas generated in KC mice using the combined Cre-loxP and CRISPR-Cas9 technology recapitulate many features of the mouse UPS model generated in KP mice by Cre-loxP alone. Therefore, similar to the KP sarcoma model, we anticipate that sarcomas generated by CRISPR-Cas9 could also be used as a platform to study lung metastasis, to investigate mechanisms of tumour response to cancer therapy and to test new imaging agents[3,40–43]. We showed that one sgRNA efficiently targeted Cas9 to both *Trp53* alleles and both on-target chromosome 17 alleles, which suggests that this system will also be useful for editing potential tumour modifying genes identified in human sarcomas to test their role during sarcomagenesis. This system can also be used to study tumour clonality, as unique indels are generated by non-homologous end joining after gene editing by Cas9. It is important to note that variation between two indels may be unique in some cases, but are certainly non-random[44]. Consequently, targeting only one locus makes it impossible to assess clonality with any degree of certainty due to the probability of generating identical sets of indels in two or more independent clones due to indel bias. By targeting several loci and generating indels across multiple loci, a unique indel barcode can be generated endogenously. To demonstrate the feasibility of this application, we employed an sgRNA with two different perfect on-target sites, one in the *Trp53* locus and another locus on chromosome 17, as a method of multiplexed generation of four unique indels per cell (one indel per targeted allele) that represents a specific and endogenous genetic barcode to follow clonality during soft tissue sarcoma development. Interestingly, the Sanger sequencing analysis of cell lines from three independent sarcomas suggests that these sarcoma cell lines were derived from a single clone. It is possible that clonal selection occurred during *in vivo* tumour evolution or during *in vitro* passaging to deplete wild-type stroma. Indeed, one sarcoma (2981) harboured a single base pair insertion in the targeted *Trp53* locus that was identified in 30 clones sequenced by Sanger sequencing, although deep sequencing of the original

tumour identified this single nucleotide insertion in less than 1% of the reads (Supplementary Table 2).

In addition, a single base pair deletion in the targeted Trp53 locus in another sarcoma (2995) detected by deep sequencing was rarely found by Sanger sequencing of the cell lines derived from the same tumour (Fig. 3a,b). The low frequencies of two indels in *Trp53* found by Sanger sequencing suggests that it may have developed from independent clones (Fig. 3a). It is also possible that tetraploidization before mutagenesis may account for the detection of more than two distinct indels. In this scenario, however, the frequency of indels generated in a tetraploid intermediate may be more evenly distributed than what we observed. Therefore, it is likely to be that the sarcoma sample may have contained a dominant clone and other rare clones. These data also suggest that the endogenous genetic barcodes generated by Cas9 can be used to study tumour clonality at different stages of tumour development. Here we demonstrate the feasibility of this approach using four genomic loci as potential endogenous barcodes. Multiplexed sgRNAs targeting more than four genomic loci will enable greater sensitivity for identifying and tracking truly clonal populations.

We also performed deep sequencing of the top ten predicted off-target sites in five primary sarcomas in KC mice following Ad-P-Cre injection. The rate of small indels in tumours was similar to the control normal muscle (Supplementary Table 3), indicating no off-target activity at these sites. Sanger sequencing of the on-target sites in exon 7 of Trp53 revealed the presence of deletions up to 281 bp. This class of indel was most commonly represented in our Sanger sequencing data (Fig. 3a). For indel classes that could be detected by deep sequencing, such as the insertion in mouse 2981 and deletion in mouse 2925, the deep sequencing was validated by Sanger sequencing. However, next-generation sequencing did not detect a class of indel, which were large deletions. For example, we found deletions up to 281 bp in size via Sanger sequencing, but these large deletions were not detected by deep sequencing of the same tumour DNA (Fig. 3a,b). Failure of targeted deep sequencing to capture these large deletions class of indels may be due to a limitation of the deep sequencing primer design, which did not adequately flank the targeted loci to allow DNA amplification in the event of a large deletion resulting from DNA repair. Therefore, to capture these larger deletions we searched for deletions in the single exonic off-target site (Celsr1) using Sanger sequencing (Supplementary Table 4). These data showed exon 2 of Celsr1 contained variants, but these were identified as SNPs outside of the Cas9 recognition site (Supplementary Fig. 3). In addition, we performed Surveyor analysis and Sanger sequencing on the remaining off-target sites. No DNA mismatches were detected in those sites, except in off-target site 4, which we also determined to be SNPs (Supplementary Fig. 4).

Other groups have utilized lentiviral delivery for somatic genome editing with the CRISPR-Cas9 system to generate primary cancers of the lung, pancreas and breast[16,18,19]. We used an adenoviral delivery system, because sarcoma induction via lentiviral expression of Cre in immunocompetent mice is inefficient apparently because of immunosurveillance[45]. Compared with intramuscular injection of an Ad-Cre into KP mice, we found that intramuscular injection of adenovirus expressing Cre and sgTrp53 into KC mice led to soft tissue sarcoma formation with similar penetrance, time to tumour onset, histologic appearance, mutational load and CNV. In the interest of further simplifying the generation of sarcomas in KC mice, we also performed intramuscular injection of a naked plasmid expressing Cre and sgTrp53. Despite being convenient and economical, intramuscular injection of naked plasmid DNA resulted in primary sarcoma formation with delayed kinetics in

only $\sim 25\%$ of mice. However, the efficiency of naked plasmid delivery to the tumour initiating cell(s) can be increased using IVE[46,47], which we hypothesized would increase the penetrance of sarcoma development[48]. Therefore, we delivered pX333-P-Cre plasmid via IVE and successfully generated tumours in 80% of KC mice (median 10.8 weeks). In KC mice infected with Ad-P-Cre, sarcomas arose over a similar time frame ranging from 7.3–15.7 weeks (median 9.6). Remarkably, we noticed that delivery of pX333-P-Cre plasmid via IVE in mice that constitutively express Cas9 (K-loxP-C mice) resulted in 100% tumour penetrance (median 7.4 weeks) with a more restricted period for tumour formation. This temporal difference between the models may be due to either a cell autonomous or non-cell autonomous mechanism. For example, the endogenous Cas9 expressed in K-loxP-C mice may increase the target population of cells that undergo both Cre-mediated recombination and Cas9-mediated genome editing that are required for tumorigenesis. Alternatively, the endogenous Cas9 expressed in K-loxP-C mice before tumour initiation may delete self-reactive T-cells that recognize Cas9 during normal mouse development. Therefore, in comparison to the KC mice where Cas9 is expressed only after Cre-mediated recombination to generate a potential neoantigen for immune surveillance, the constitutive expression of Cas9 may limit immune surveillance in the K-loxP-C model. Regardless of the mechanism, these data establish *in vivo* plasmid DNA electroporation as an efficient and facile method of generating sarcomas via CRISPR-Cas9 technology.

We also successfully generated a sarcoma model of MPNST using CRISPR-Cas9 mutagenesis to knockout *Nf1* and *Trp53* in wild-type mice. Generating this MPNST model only required the time to generate the virus (8 weeks), which is much faster than targeting alleles in ES cells, generating mice with the targeted alleles, crossing the multiple mutant alleles together, and genotyping the mice to generate the desired experimental mice. More importantly, this approach to generating MPNSTs and other sarcomas in wild-type mice gives great flexibility for studying the function of additional genetic mutations without the extra time, effort and expense required for additional crosses.

The mutational landscapes of GEMMs of lung adenocarcinoma were recently reported to have >10-fold lower mutational loads $(0.02–0.14 \, \text{MB}^{-1})$ compared with human lung cancer from non-smokers $(1.97 \, \text{MB}^{-1})$[49]. We performed whole exome sequencing to profile the mutational load and CNV in the KP primary sarcoma models initiated by Cre alone or Cre and CRISPR-Cas9. We found the mutational load in Cre-mediated KP tumours $(0.09 \, \text{MB}^{-1})$ was comparable to that of CRISPR-mediated KP tumours $(0.07 \, \text{MB}^{-1})$ (Fig. 6c). Moreover, the CNV frequency was similar between sarcomas generated with Cre-only $(0.05 \, \text{MB}^{-1})$ or Cre and CRISPR-Cas9 $(0.05 \, \text{MB}^{-1})$ (Fig. 6d). Interestingly, the mutational load in the CRISPR-mediated $(0.07 \, \text{MB}^{-1})$ and Cre-mediated KP sarcomas $(0.09 \, \text{MB}^{-1})$ was similar to the mutational load in KP lung adenocarcinomas $(0.07 \, \text{MB}^{-1})$[49]. These mutational profiles further demonstrate the similarity between Cre-initiated and CRISPR-Cas9-initiated sarcomas on a genomic level and provide additional evidence beyond the sequencing of potential off-target sites that sarcomas generated by CRISPR-Cas9 technology do not harbour a significant increase in mutations.

In conclusion, we have generated and characterized a combined Cre-loxP, CRISPR-Cas9 model of primary UPS. These sarcomas are comparable to conventional GEMMs of cancer with similar histology, growth kinetics and mutational profiles (Fig. 6). Furthermore, we showed that the CRISPR-Cas9 system can be used to generate autochthonous MPNSTs in wild-type mice. As these models are less expensive and time consuming than traditional GEMMs of sarcoma, they should enable a larger

number of investigators to employ primary mouse models for cancer research.

## Methods

**Mice.** $Kras^{LSL-G12D/+}$ mice were provided by T. Jacks (Massachusetts Institute of Technology). $Rosa26^{LSL-Cas9-EGFP/+}$ mice were provided by F. Zhang (Massachusetts Institute of Technology). 129/SvJ wild-type mice were purchased from the Jackson Laboratory. $Rosa26^{loxP-Cas9-EGFP/+}$ mice were generated by crossing $Rosa26^{LSL-Cas9-EGFP/+}$ mice with Meox2Cre mice and then crossing with C57BL/6J wild-type mice. For the allograft study, male athymic nude (nu/nu) mice (5 to 6 weeks old) were purchased from Taconic Biosciences and maintained in Duke University's accredited animal facility. For all other experiments, both male and female mice were used. MEFs transduced with $1 \times 10^6$ virus particles or transiently transfected with plasmid DNA were injected into the hind limb of the nude mice. Virus was prepared by mixing 25 µl of Ad-sgTrp53-cre ($6 \times 10^7$ pfu µl$^{-1}$) or Ad-CMV-EGFP ($4 \times 10^7$ pfu µl$^{-1}$) with 600 µl minimal essential medium (Sigma-Aldrich, M4655) or mixing 125 µl of Ad-sgNf1-sgTrp53-Cas9 ($4 \times 10^6$ pfu µl$^{-1}$) with 500 µl minimal essential medium. Three microlitres of 2 M $CaCl_2$ was added to each virus prep, mixed and incubated for 15 min at room temperature before injection into either the skeletal muscle (50 µl) or the sciatic nerve (25 µl) after the nerve was exposed by an incision in the surrounding muscle[3,25]. All animal studies were performed in accordance with protocols approved by the Duke University Institutional Animal Care and Use Committee.

**MEF isolation.** MEFs were generated from E13.5 to E14.5 embryos following standard procedures. Passage 2–5 MEFs were infected with adenovirus using the stated multiplicity of infection or transiently transfected with plasmids using TransIT-2020 reagent (Mirus Bio LLC, MIR5400) according to the manufacturer's instructions.

**Cell culture.** NIH-3T3 cell lines were purchased from ATCC (CRL-1658) and cultured in DMEM medium (Thermo Fisher Scientific, 11965092) supplemented with 10% fetal bovine serum and 1% antibiotic–antimycotic (Thermo Fisher Scientific, 10091148), and incubated at 37 °C with 5% $CO_2$ in a humidified cell-culture incubator. Tumour cells were isolated and expanded *in vitro* from harvested tumours using standard methods[43]. Briefly, tumour tissues were minced in the cell-culture hood and then they were digested by dissociation buffer in $1 \times$ PBS (Thermo Fisher Scientific, 14040133) containing collagenase type IV (5 mg ml$^{-1}$, Thermo Fisher Scientific, 17104-019), dispase (1.3 mg ml$^{-1}$, Thermo Fisher Scientific, 17105-041) as well as trypsin (0.05%, Thermo Fisher Scientific, 25200056) for about 1 h at 37 °C. Cells were washed with $1 \times$ PBS (Thermo Fisher Scientific, 10010023) and filtered using a 40 µm sieve (Corning, 431750), and cultured for at least four passages before being used for experiments.

**Plasmids and adenoviral vectors.** The Ad-sgTrp53-Cre adenoviral vector was constructed as follows. The pX333 vector was kindly provided by A. Ventura (Memorial Sloan Kettering Cancer Center). The Cas9 gene in pX333 was replaced by Cre recombinase (Addgene 60229) using AgeI and EcoRI sites by standard cloning methods. For cloning pX333-sgTrp53-Cre, the pX333-Cre vector was digested with BsaI enzyme and ligated to annealed sgRNA oligonucleotides (Supplementary Table 5) targeting mouse *Trp53* exon 7. In the final step, pX333-sgTrp53-Cre vector was cloned into the Ad5 adenoviral shuttle vector, which was kindly provided by A. Ventura (Memorial Sloan Kettering Cancer Center) using XhoI and EcoRI sites. The pcDNA3-EGFP vector was purchased from Addgene (13031). Restriction enzymes and T4 ligase were purchased from New England Biolabs. The pX333-sgNF1-sgTrp53-Cas9 adenoviral vector was constructed using a similar approach. Recombinant adenoviruses were generated by Viraquest Inc.

**Molecular analysis of recombination.** Recombination of the mutant *Kras* allele and the *Cas9* allele were assessed by PCR using AccuPrime Taq HiFi DNA polymerase (Thermo Fisher Scientific, 12346-086)[10,50] and utilized primers in Supplementary Table 5.

**Tumour analysis.** Harvested tissues were fixed in 4% formalin and paraffin-embedded. Immunohistochemistry was performed on 5 µm sections with the ABC kit (Vector Laboratories, PK-7200) with antibodies to S100 (Dako, GA50461-2), MyoD (Dako, M351201-2) and myogenin (Dako, IR06761-2). Sections were developed with 3,3′-diaminobenzidine and counterstained with haematoxylin (Sigma-Aldrich, H3136). Haematoxylin and eosin staining was performed using standard methods. All tissue sections were examined by a sarcoma pathologist (D.M. Cardona).

**Frozen tissue processing.** The gastrocnemius muscle was harvested from sacrificed mice and immersed in a 4% paraformaldehyde/PBS (Affymetrix, 19943) solution overnight at 4 °C. The tissue was washed in PBS twice for 5 min before immersing in a 30% sucrose/PBS solution overnight at 4 °C. The tissue was then blotted and placed into Tissue-Tek OCT (Sakura Finetek USA, Inc., 4583)

compound in embedding molds before snap-freezing in a dry ice and 70% ethanol slurry. Tissue blocks were stored at $-80$ °C until sectioned at 10 µm using a cryostat. Slides were washed once in PBS, mounted with ProLong Diamond Antifade Mountant with DAPI (Thermo Fisher Scientific, P36962) and imaged using fluorescence microscopy. GFP-positive cells were counted by ImageJ software.

**Genomic DNA isolation and indel analysis.** Genomic DNA was isolated from both cells and tissues using DNeasy Blood & Tissue Kit (Qiagen, 69504) following the manufacturer's protocol. PCR amplification of mouse *Trp53* targeted exon 7 was performed with specific primers using AccuPrime Taq HiFi DNA polymerase. PCR amplification of mouse *Nf1* targeted exon 7 was performed with specific primers. PCR reaction was subjected to a re-annealing process to enable heteroduplex formation in a thermocycler[37]. CRISPR-Cas9-induced indels were detected using Surveyor Mutation Detection Kits (IDT, 706021) following the recommended protocol. Homoduplex and cleaved heteroduplex products were visualized with ultraviolet illumination after gel electrophoresis in 2% Tris-borate-EDTA agarose gels, or 4–20% Tris-borate-EDTA polyacrylamide gels.

**Molecular analysis of Kras$^{G12D}$ mutation.** PCR amplification of mouse wild-type Kras allele and the Kras$^{G12D}$ mutant allele was performed with specific primers (Supplementary Table 1). The PCR product was either directly or cloned into the pCR4-TOPO TA vector (Thermo Fisher Scientific, K457501) for Sanger sequencing. A total of five clones from each vector were Sanger sequenced.

**Western blot analysis of p53 and Nf1 knockout.** NIH-3T3, KP tumour and MPNST tumour cell lines were treated with 0.5 µg ml$^{-1}$ doxorubicin (Sigma-Aldrich, D1515-10MG) for 8 h before sample harvest. Samples were lysed in RIPA buffer for 30 min on ice (Sigma-Aldrich, R0278) then centrifuged at 10,000 g for 20 min at 4 °C. Lysate supernatant was separated from debris into a new tube and protein concentration was determined by BCA assay (Pierce, 23225). The lysate was boiled in $4 \times$ sample buffer (Thermo Fisher Scientific, 84788) at 95 °C for 5 min, then cooled to room temperature before loading in a 3–8% Tris-Acetate protein gel (Thermo Fisher Scientific, EA03785BOX). Samples were electrophoresed at 150 V for 80 min after transfer to nitrocellulose. Total protein was determined using REVERT total protein stain (Li-Cor Biosciences, P/N 926-11011) and an Odyssey CLx (Li-Cor Biosciences, 9194), then washed and destained before blocking in 5% non-fat dry milk in tris-buffered saline (TBS, Corning, 46-012-CM). Next, membranes were incubated overnight at 4 °C with primary antibodies diluted in TBS-T (0.1% Tween-20): p53, 1:250 dilution (1C12, Cell Signaling Technology); Nf1, 1:500 dilution (A300-A140M, Bethyl Laboratories). Membranes were washed three times in TBS-T for 5 min before secondary antibody incubation with goat anti-rabbit IRDye800 (Li-Cor Biosciences, P/N 925-32211) and goat anti-mouse IRDye680 (Li-Cor Biosciences, P/N 925-68070) both at 1:10,000 dilutions in TBS-T for 1 h at room temperature. The membranes were washed three times in TBS-T for 5 min and imaged using an Odyssey CLx (Li-Cor Biosciences). Image analysis for normalization and quantification were performed using Image Studio (Version 5.2, Li-Cor Biosciences, P/N 9140-500). Full blots with protein ladders (Li-Cor Biosciences, P/N 928-60000) are shown in Supplementary Fig. 7.

**Clonality study.** Cells were disassociated from four independent sarcomas and passaged at least five times. Genomic DNA was isolated from each tumour cell line. *Trp53* exon 7 was amplified by PCR and cloned into the pCR4-TOPO TA vector (Thermo Fisher Scientific). Thirty clones from each vector were Sanger sequenced. On-target chromosome 17 site was amplified by PCR using genomic DNA isolated from each tumour cell line and cloned into pCR4-TOPO TA vector. Ten clones from each vector were Sanger sequenced for the on-target chromosome 17 site.

**Whole exome sequencing.** Genomic DNA was sheared to 250 bp using the Covaris S2 platform. To increase the throughput of exome library preparation, a custom 96-well barcode system was assigned based on the Illumina sequencing system, whereby a separate sequencing read is used to identify the barcode of individual sequencing libraries. Custom 8-mer barcodes compatible with the 3-read Illumina Hiseq v2 platform were designed enforcing an edit distance of $\geq 3$ to greatly minimize the possibility of sample cross-contamination during sequencing. Additional measures were applied to ensure optimal base diversity at each sequencing cycle.

Oligonucleotides were ordered with standard Illumina barcode modifications of 5′-phosphorylation, a phosphorothioate bond between the last two bases on the 3′-end, and HPLC purification. Oligonucleotides were annealed by standard protocols and diluted to a 15 M working concentration in elution buffer (EB). Pre-capture libraries were prepared with standard library preparation protocols using the KAPA Hyper kit (Kapa Biosystems) and then pooled at equal volume and subjected to exome-capture using the Agilent Mouse All Exon kits, followed by high-throughput sequencing on the Illumina Hiseq 2500 platform.

Reads in fastq format were pre-processed with GATK[51] version 3.2 to remove Illumina adapter sequences and phred-scaled base qualities of ten and below. After

GATK processing, reads were mapped to mm10 using Burrows-Wheeler Aligner[52] version 0.7.7 with the mem algorithm. Reads were sorted with Novoalign V2.08.03 novosort. PCR/optical duplicates were marked by Picard. Base-quality recalibration and localized indel realignment was performed using GATK[53].

Joint variant calling was performed to identify single nucleotide variant and indels using GATK Haplotype Caller[53] by first assembling plausible haplotypes to reduce alignment errors and to improve the accuracy of variant calling. Somatic variants were identified using MuTect[54] by comparing tumours with matched normal. Variants were annotated using SnpEff[55].

Variant filtering was performed by first taking the intersection of the variants from both the variant callers (MuTect and Haplotype caller) to identify high-quality somatic variants. Additional filtering was performed to retain variants with good coverage ($\geq 15$ reads), allele frequency ($\geq 0.2$), exonic variants predicted to change the amino acid sequence (missense, frameshift, stop gain and splice site variants). Mutation load per MB was calculated by dividing the count of the filtered somatic variants by the size of the exome target region (49.5 MB).

PCR amplification of each somatic variant was performed with specific primers using AccuPrime Taq HiFi DNA polymerase (Thermo Fisher Scientific). The PCR product was either used directly or cloned into the pCR4-TOPO TA vector (Thermo Fisher Scientific) for Sanger sequencing. A total of 10 or 30 clones from each vector were Sanger sequenced. Total 38 somatic mutation calls were validated independently, which implied 84.2% accuracy of the calling method. Validated mutations for each tumour tissue are listed in Supplementary Table 6.

**Deep sequencing.** The sgTrp53 target and the top ten candidate off-target sites were examined in five sarcomas generated by Ad-P-Cre in KC mice by deep sequencing following a protocol described previously[11]. PCR of genomic DNA from 5 sarcomas and the normal muscle from the same mouse was carried out using primers spanning the on-target *Trp53* region, on-target chromosome 17 site and 10 off-target sites (Supplementary Tables 2 and 3). PCR was carried out to add Illumina flowcell binding sequences and experiment-specific barcodes on the 5′-end of the primer sequence (Supplementary Table 5). All PCR products were pooled together and sequenced on an Illumina MiSeq instrument. Indel analysis was performed as previously described using default CRISPResso settings and a 20 bp window[11,56]. Briefly, 3′-ends of each paired-end reads are overlapped, because PCR products were <250 bp in length. This overlap was applied to generate a consensus PCR amplicon for each paired-end read by single ungapped alignment (parameterized to score each match as 5 and each mismatch as −4). Trimmed fragments were aligned to the reference genome. Then, each of the aligning fragments was aligned to the reference genome sequence of the PCR product with a global affine alignment with the following parameterization: match = 5, mismatch = −4, gap open = −5 and gap exten = −2. If the fragment did not have greater than 60% identity with original expected PCR amplicon or it had a top-scoring BLAT alignment that was different than the expected PCR product, this fragment was discard from downstream analysis. Alignments were trimmed to a 20 bp window centred three base pairs 5′ of the protospacer-adjacent motif (PAM), the predicted site of Cas9 nuclease activity. Next, indel statistics were further gathered from windows separately for each sample by counting gaps in the query and subject sequences of the resulting truncated alignments and tabulating numbers of fragments having any indels in the windows.

**In vivo electroporation.** After mice were anaesthetized, 50 μg of naked DNA plasmid diluted in sterile saline was injected into the gastrocnemius using a 31-gauge insulin syringe. A pair of needle electrodes with a 5 mm gap was inserted into the muscle to encompass the DNA injection site, and electric pulses were delivered using an electric pulse generator (Electro Square Porator ECM830; BTX, San Diego, CA). Three 100 V pulses followed by three additional 100 V pulses of the opposite polarity were administered to each injection site at a rate of 1 pulse per 50 ms with each pulse being 200 ms in duration.

**Statistical analyses.** Results are presented as means ± s.e.m. unless otherwise indicated. Before analysis, all data were displayed graphically to determine whether parametric or non-parametric tests should be used. Two-tailed Student's *t*-test was performed to compare the means of two groups. For tumour kinetic studies, Kaplan-Meier analysis was performed with the log-rank test for statistical significance. Significance was assumed at $P < 0.05$. All calculations were performed using Prism 6 (GraphPad).

**Data availability.** All data generated or analysed during this study are included in this published article and its Supplementary Information files. The data sets generated during and/or analysed during the current study are available from the corresponding author on reasonable request. All sequencing data are deposited at the European Genome-phenome Archive with accession number: EGAS00001002459.

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

## Acknowledgements

We thank David Ousterout for suggestions on designing the sgRNA to Trp53. We also thank Andrea Ventura and Danilo Maddalo for sharing plasmids and helpful advice to generate vectors to utilize the CRISPR-Cas9 system. This work was supported by the National Cancer Institute of the US NIH under award numbers F30CA206424 (M.C.), T32GM007171 (M.C. and M.J.W.) and R35CA197616 (D.G.K.), and by the Department of Defense under award number W81XWH-14-1-0067 (D.G.K.).

## Author contributions

J.H., M.C., M.J.W. and D.G.K. designed experiments. J.H., M.C., M.J.W., H.-C.K., E.S.X., A.W., L.L., Y.M., Y.M.M., D.V.M., O.M.L., J.N.R.-H. and R.D.D. performed experiments. J.H., M.C. and E.S.X. performed IVE. W.C.E. and R.D.D. performed sciatic nerve injections. M.C., Y.M. and R.D.D. performed histology and D.M.C. analysed all IHC sections. A.R. and S.S.D. performed and analysed whole exome sequencing. J.H., M.C., H.-C.K. and C.E.N. performed and analysed deep sequencing. C.A.G. and D.G.K. provided critical advice for the manuscript. J.H., M.C., M.J.W. and D.G.K. drafted the first version of the manuscript. All authors edited the manuscript.

## Additional information

**Competing interests:** C.A.G. is a founder and scientific advisor to Element Genomics. The remaining authors declare no competing financial interests.

**Publisher's note**: 

