## [Peer review file · Nature Communications]

Reviewers' comments:

Reviewer #1 (Remarks to the Author):

The article reports on a series of experiments that tested the efficiency of in vivo application of CRISPR/Cas9 technology to generate mouse genetic models of sarcoma. Such an effort has not previously been reported outside of CRISPR-mediated translocations that drove ARMS. Similar applications of this technology may render the generation of new mouse models of sarcoma much more available to additional investigators around the world.

The focus of the article was more of a quality assessment of the new method of engineering these models than it was a discovery of new biology. The validating comparison of these new models was therefore prior mouse models, generated with previously available technology. No validation of these models against human sarcomas was provided. This is primarily pertinent to the primary focus of the article on Kras-G12D-expression/Trp53-loss as a means to model UPS. Activating mutations in Kras are not especially prominent in UPSs in humans, which has been a criticism of the prior model to which these CRISPR-mediated models were compared. Nonetheless, as a means of evaluating this new approach, comparison to former GEMMs seems fair.

My only other criticism is that the assessments of clonality were primarily performed on derivative cell lines, which certainly undergo an additional level of ex vivo clonal selection that supersedes the selection intrinsic to tumorigenesis. The authors appropriately acknowledge this limitation in the Discussion section.

Overall, the work is convincing, reproducible, statistically valid, and will contribute meaningfully to the field of sarcoma mouse modeling research, as a new method of model generation. The validated engineering feats achieved in these experiments include the generation of three new mouse models of soft-tissue sarcoma, one of which utilizes no viral particles, but only electroporation to deliver plasmids. The experiments are all carefully documented and reasonably done.

Reviewer #2 (Remarks to the Author):

Huang and colleagues describe CRISPR/Cas9-based approaches for somatic induction of sarcomas in mice. This includes (1) a mouse model of undifferentiated sarcoma using KrasLSL-G12D;RosaLSL-Cas9-EGFP mice in combination with adenoviral delivery of Cre and a Trp53-targeting sgRNA; (2) a similar model, but using electroporation for Cre and sgRNA delivery; (3) a mouse model of primary malignant nerve sheath tumour by combinatorial targeting of Nf1 and Trp53 in wild-type mice using adenoviral sgRNA delivery.

The authors build on their long-standing expertise in modelling sarcoma, including the previous development of somatic gene activation/inactivation using adenoviral Cre-delivery. They systematically compare their CRISPR-based somatic GEMM to previous GEMM. Overall, the paper is well written and the experiments are well presented. The approaches and phenotypes are robust and the protocols will certainly be useful to others in the field for rapid sarcoma modelling in mice.

The weaknesses of the manuscript are that it is purely methods driven and that the innovative potential is relatively low because:

- (1) The vector delivery approaches (Adenovirus, Electroporation) have been developed and tested before
- (2) Applications other than simple gene editing have not been explored (E.g. chromosome engineering, screening etc)
- (3) Genes other than known sarcoma tumor suppressors have not been studied, meaning that (a) there are no new biological insights and (b) it is unclear whether the method is robust enough for cancer induction if lower-penetrance genes are targeted. There is no clear conclusion, whether the

method can be extended to test other genes (see also specific comment 1).

Specific comments:

1. It would be useful to present the efficiency of the different delivery approaches (viral and electroporation-based). What percentage of different cells types is transduced/transfected upon injection? I acknowledge that the delivery approaches have been described before and that there might be such data in the literature. Nevertheless, I would find it helpful if these data were presented in the supplement. The delivery efficiency might be an important parameter when considering to extend the method to the study of genes with lower biological impact (and lower tumour-inducing capacity) than the ones tested (p53, Nf1).

2. I don't agree with the assumption that NGS cannot detect larger deletions (because of the size restrictions of sequencing on Illumina MiSeq) and that Sanger sequencing is better at doing this. Large deletions need to be PCR-amplified for both methods and by definition a large deletion will give a smaller band (which can then of course be sequenced on MiSeq). It is just about using different primer sets for multiple screening PCRs. In addition, for the systematic analysis of off-target effects, NGS is the preferred method, as it is far more sensitive than capillary sequencing. Sensitivity might matter because off-target editing is less efficient than on-target editing (and thus off-target mutations might be subclonal). It is unclear why only part of the off-target analyses were performed using NGS.

3. Minor: The interpretation that more than 2 different CRISPR-induced mutations in one cancer sample reflect oligoclonality can be correct, but is not the only possibility. It is also possible that tumours underwent tetraploidization before mutagenesis. There are examples for this in the literature.

4. I am not convinced that the method presented is appropriate for barcode generation and tracking experiments. Given that the majority of CRISPR-induced mutations are 1bp deletions with high probability to occur at specific positions at the target site, it is impossible to predict relatedness of two cell clones based on the same 1bp mutation. This is only possible, if patterns of mutations are analysed in a context of CRISPR-based multiplexed sgRNA delivery targeting multiple genes.

Responses to Reviewers

We thank the reviewers for their thoughtful comments, suggestions, and review of our manuscript. We have performed additional experiments and revised the text of the manuscript to address every reviewer comment. Our point-by-point responses and revisions are written below in blue type under the corresponding reviewer comment in black italicized type.

Reviewer #1 (Remarks to the Author):

The article reports on a series of experiments that tested the efficiency of in vivo application of CRISPR/Cas9 technology to generate mouse genetic models of sarcoma. Such an effort has not previously been reported outside of CRISPR-mediated translocations that drove ARMS. Similar applications of this technology may render the generation of new mouse models of sarcoma much more available to additional investigators around the world.

The focus of the article was more of a quality assessment of the new method of engineering these models than it was a discovery of new biology. The validating comparison of these new models was therefore prior mouse models, generated with previously available technology. No validation of these models against human sarcomas was provided. This is primarily pertinent to the primary focus of the article on Kras-G12D-expression/Trp53-loss as a means to model UPS. Activating mutations in Kras are not especially prominent in UPSs in humans, which has been a criticism of the prior model to which these CRISPR-mediated models were compared. Nonetheless, as a means of evaluating this new approach, comparison to former GEMMs seems fair.

We agree completely with Reviewer #1's assessment that the focus and scope of this manuscript is on comparing two methods of engineering mouse models of sarcoma. Although CRISPR/Cas9 technology is now widely used, mouse modeling is an area where it will complement, and in some cases replace, existing technologies such as site-specific recombinases (SSR). A comparison between CRISPR and SSR systems for generating mouse models of cancer was the primary focus of this manuscript. A secondary focus was demonstrating the ability to rapidly generate multiple subtypes of sarcoma in mice, using both virus and in vivo electroporation methods of delivery.

My only other criticism is that the assessments of clonality were primarily performed on derivative cell lines, which certainly undergo an additional level of ex vivo clonal selection that supersedes the selection intrinsic to tumorigenesis. The authors appropriately acknowledge this limitation in the Discussion section.

We agree with Reviewer #1 that assessment of clonality in derivative cell lines does create a level of ex vivo clonal selection as we acknowledged in the Discussion. We tried to address this limitation experimentally by performing deep sequencing of primary sarcomas in Figure 3B and 3D to complement the experiments in the cell lines. To better highlight this for readers we have modified the text in the Results section of the revised manuscript:

Results, page 8 (changes highlighted in yellow)

These data demonstrate that the high efficiency of genome editing by sgTrp53 is not restricted to genes required for transformation, suggesting that the CRISPR/Cas9 system could be utilized

to target other genes to test their ability to modify sarcoma development and other phenotypes. Because clonal analysis of derivative cell lines may introduce an additional level of *ex vivo* clonal selection, we also analyzed primary sarcomas (n = 5) by deep sequencing of *Trp53* and the on-target chromosome 17 sites.

Overall, the work is convincing, reproducible, statistically valid, and will contribute meaningfully to the field of sarcoma mouse modeling research, as a new method of model generation. The validated engineering feats achieved in these experiments include the generation of three new mouse models of soft-tissue sarcoma, one of which utilizes no viral particles, but only electroporation to deliver plasmids. The experiments are all carefully documented and reasonably done.

We thank Reviewer #1 for finding our manuscript “convincing, reproducible, statistically valid, and will contribute meaningfully to the field.”

Reviewer #2 (Remarks to the Author):

*Huang and colleagues describe CRISPR/Cas9-based approaches for somatic induction of sarcomas in mice. This includes (1) a mouse model of undifferentiated sarcoma using *Kras^{LSL-G12D};Rosa^{LSL-Cas9-EGFP}* mice in combination with adenoviral delivery of Cre and a *Trp53*-targeting sgRNA; (2) a similar model, but using electroporation for Cre and sgRNA delivery; (3) a mouse model of primary malignant nerve sheath tumour by combinatorial targeting of *Nf1* and *Trp53* in wild-type mice using adenoviral sgRNA delivery. The authors build on their long-standing expertise in modelling sarcoma, including the previous development of somatic gene activation/inactivation using adenoviral Cre-delivery. They systematically compare their CRISPR-based somatic GEMM to previous GEMM. Overall, the paper is well written and the experiments are well presented. The approaches and phenotypes are robust and the protocols will certainly be useful to others in the field for rapid sarcoma modelling in mice.*

We thank Reviewer #2 for recognizing that our “approaches and phenotypes are robust” and finding that our protocols will “certainly be useful to others in the field.”

The weaknesses of the manuscript are that it is purely methods driven and that the innovative potential is relatively low because:

(1) The vector delivery approaches (Adenovirus, Electroporation) have been developed and tested before
We acknowledge that this manuscript is heavily methods driven and relies on previously developed methods of vector delivery (Adenovirus, Electroporation) and gene editing (CRISPR/Cas9). The first papers demonstrating successful application of CRISPR/Cas9 technology to mouse modeling did use a similar approach. These seminal papers pioneered *in vivo* applications of CRISPR/Cas9 technology in mice¹⁻⁴. Subsequent papers by other groups applied these same techniques for other cancers as we noted in the manuscript⁵⁻⁸. Here we expand the scope of cancers that can be modeled in mice using CRISPR/Cas9 by generating primary sarcomas for the first time. In addition, this research is novel and interesting to the broader CRISPR and mouse modeling communities because we used our mouse models as a system to comprehensively compare the sarcomas generated using either conventional SSR technology or CRISPR/Cas9 technology. Recently, a report by McFadden et al.⁹ compared the mutational

landscape of human and mouse cancers and showed that human and mouse cancers have different mutational profiles. This finding raised important considerations for downstream applications of mouse models of cancer, for which mutational load may be a confounding factor. Similarly, we think the impact of our research showing that SSR and CRISPR/Cas9 technologies generate very similar tumors in mice is important to evaluate past GEMM experiments and plan future experiments as CRISPR/Cas9 inevitably becomes more widespread in cancer mouse modeling.

(2) Applications other than simple gene editing have not been explored (E.g. chromosome engineering, screening etc)

We acknowledge that we use simple knockout gene editing as the primary application of CRISPR/Cas9 technology. We also use CRISPR/Cas9 to examine tumor clonality by using a sgRNA that targets two sites in the genome: one site in Trp53 to generate a Trp53 knockout, and another site in an intergenic region of chromosome 17. The unique permutations of indels generated at potentially four sites in the genome can be used to examine clonality. We agree with Reviewer #2 that this technology can be applied for more sophisticated gene editing, such as chromosomal translocations. Indeed we have recently embarked on a project that is utilizing this approach. However, the data are too preliminary to include in this manuscript and we feel that this work is outside the scope of the current manuscript.

(3) Genes other than known sarcoma tumor suppressors have not been studied, meaning that (a) there are no new biological insights and (b) it is unclear whether the method is robust enough for cancer induction if lower-penetrance genes are targeted. There is no clear conclusion, whether the method can be extended to test other genes (see also specific comment 1).

We agree that using CRISPR to rapidly test lower-penetrance genes is an important application of the system we describe in our manuscript. Indeed, selecting a sgRNA that targets both Trp53 and a region of chromosome 17 that is irrelevant for sarcoma induction, we have demonstrated that our system can not only introduce mutations into known tumor suppressors like Trp53, but also robustly mutates other sites of DNA that are not required for sarcomagenesis. Therefore, the results we describe in Figure 3C and 3D with editing chromosome 17 demonstrate that the system can be extended to loci other than established tumor suppressor genes. While we agree with Reviewer #2 that we have not used this system to test the impact of lower-penetrance genes on sarcoma development, we believe this experiment is not necessary to support the conclusions of the current manuscript and is outside the scope of this manuscript. Instead, using our system to study the role of other genes in sarcoma development is planned in future studies.

Specific comments:

1. It would be useful to present the efficiency of the different delivery approaches (viral and electroporation-based). What percentage of different cells types is transduced/transfected upon injection? I acknowledge that the delivery approaches have been described before and that there might be such data in the literature. Nevertheless, I would find it helpful if these data were presented in the supplement. The delivery efficiency might be an important parameter when considering to extend the method to the study of genes with lower biological impact (and lower tumour-inducing capacity) than the ones tested (p53, Nf1).

We thank the reviewer for suggesting this experiment to quantify the differential efficiency between adenoviral and *in vivo* electroporation (IVE)-based approaches, which will be important for future applications of either IVE or adenovirus for the reasons mentioned by reviewer #2. We have therefore performed a new experiment to address this question by comparing the efficacy of GFP delivery between adenovirus and electroporation of a GFP plasmid. We found that IVE has an approximately 38-fold higher delivery efficiency compared to adenovirus based on GFP-expression 3-days post-delivery into the gastrocnemius muscle of mice. These data are now presented in Figure S5 of the manuscript and incorporated into the revised manuscript in the Results and Materials and Methods sections.

Figure S5. Comparison of the efficiency of GFP expression in muscle after *in vivo* electroporation (IVE) and adenovirus delivery. (A) Wild type mice received intramuscular delivery of either an adenovirus expressing EGFP (n = 3) or naked plasmid pcDNA3-EGFP with IVE (n = 3). Skeletal muscle from the site of injection or IVE was analyzed for GFP expression 3 days later. (B) Fluorescence microscopy was performed and representative images are shown. (C) Quantification of the number of GFP-positive cells in skeletal muscle in a section from three different mice that received intramuscular injection of adenovirus or IVE. Scale bars = 200 μ m.

Results, page 10 (changes highlighted in yellow)

The time frame for tumor development via naked plasmid injection was both delayed and less efficient compared to Ad-P-Cre virus delivery. However, electroporation of plasmid efficiently generated tumors (80% penetrance after median 10.8 weeks, Fig. 4) with similar kinetics and histology as delivery of Ad-P-Cre (median 9.6 weeks, Fig. 2B, 2D). *In vivo* electroporation of

naked plasmid pcDNA3-EGFP was approximately 38 times more efficient for EGFP delivery to muscle fibers compared to adenovirus delivery of EGFP, but this increased efficiency did not affect the penetrance or the kinetics of sarcoma formation (Supplementary Fig. 5).

Materials and Methods, page 21 (changes highlighted in yellow)

Frozen Tissue Processing. The gastrocnemius muscle was harvested from sacrificed mice and immersed in a 4% paraformaldehyde/PBS (Affymetrix, 19943) solution overnight at 4°C. The tissue was washed in PBS twice for 5 minutes before immersing in a 30% sucrose/PBS solution overnight at 4°C. The tissue was then blotted and placed into Tissue-Tek OCT (Sakura Finetek USA Inc, 4583) compound in embedding molds before snap-freezing in a dry ice and 70% ethanol slurry. Tissue blocks were stored at -80°C until sectioned at 10 µm using a cryostat. Slides were washed once in PBS, mounted with ProLong Diamond Antifade Mountant with DAPI (Thermo Fisher Scientific, P36962), and imaged using fluorescence microscopy. GFP-positive cells were counted by ImageJ software.

2. I don't agree with the assumption that NGS cannot detect larger deletions (because of the size restrictions of sequencing on Illumina Miseq) and that Sanger sequencing is better at doing this. Large deletions need to be PCR-amplified for both methods and by definition a large deletion will give a smaller band (which can then of course be sequenced on MiSeq). It is just about using different primer sets for multiple screening PCRs. In addition, for the systematic analysis of off-target effects, NGS is the preferred method, as it is far more sensitive than capillary sequencing. Sensitivity might matter because off-target editing is less efficient than on-target editing (and thus off-target mutations might be subclonal). It is unclear why only part of the off-target analyses were performed using NGS.

We agree with this criticism that NGS has higher sensitivity than Sanger sequencing for detection of off-target mutations. We performed NGS for all samples at both the *Trp53* and chromosome 17 sgRNA on-target sites, but because of the primers used for NGS, larger deletions may not have been captured. The motivation for performing Sanger sequencing on our samples was to capture these large deletions, which were found at both the *Trp53* and chromosome 17 sgRNA on-target loci in our tumor samples. We apologize that this was not clearly stated. By using both NGS and Sanger sequencing we found no significant off-target mutations and confirmed that some mutations found using Sanger sequencing were not detected using NGS (Table S3). As Reviewer #2 stated, this does not reflect a low sensitivity with NGS, but rather a limitation of the deep sequencing primer design. We have clarified this point in the revised text:

Results, page 8 (changes highlighted in yellow)

Several deletions in the *Trp53* and on-target chromosome 17 sites identified by Sanger sequencing of the cell lines were too large to be detected by deep sequencing. This limitation of our deep sequencing was most likely due to primer design, which failed to amplify the targeted loci in the presence of a large indel that may have eliminated a complementary primer sequence required for PCR amplification.

Discussion, page 15 (changes highlighted in yellow)

Sanger sequencing of the on-target sites in exon 7 of *Trp53* revealed the presence of deletions

up to 281 bp. This class of indel was most commonly represented in our Sanger sequencing data (Fig. 3A). For indel classes that could be detected by deep sequencing, such as the insertion in mouse 2981 and deletion in mouse 2925, the deep sequencing was validated by Sanger sequencing. However, NGS did not detect a class of indel, which were large deletions. For example, we found deletions up to 281 bp in size via Sanger sequencing, but these large deletions were not detected by deep sequencing of the same tumor DNA (Fig. 3A, B). Failure of targeted deep sequencing to capture these large deletions class of indels may be due to a limitation of the deep sequencing primer design, which did not adequately flank the targeted loci to allow DNA amplification in the event of a large deletion resulting from DNA repair. Therefore, to capture these larger deletions we searched for deletions in the single exonic off-target site (*Celsr1*) using Sanger sequencing (Supplementary Table 4). These data showed exon 2 of *Celsr1* contained variants, but these were identified as SNPs outside of the Cas9 recognition site (Supplementary Fig. 3).

3. Minor: The interpretation that more than 2 different CRISPR-induced mutations in one cancer sample reflect oligoclonality can be correct, but is not the only possibility. It is also possible that tumours underwent tetraploidization before mutagenesis. There are examples for this in the literature.
Thank you for suggesting this alternative explanation. Tetraploidization before mutagenesis is a possibility for our observations in tumor 2995. We have addressed this in the revised Results and Discussion sections:

Results, page 7, (changes highlighted in yellow)

Interestingly, four distinct indels in the *Trp53* site were detected in one cell line (2995), suggesting that this sarcoma cell line either contained at least two clones or arose because of tetraploidization prior to mutagenesis¹⁰⁻¹².

Results, page 9, (changes highlighted in yellow)

However, three distinct indels in the on-target chromosome 17 site were detected in one tumor (2995). This finding further indicated that this sarcoma contained either multiple clones, or underwent tetraploidization before mutagenesis.

Discussion, page 14 (changes highlighted in yellow)

In addition, a single base pair deletion in the targeted *Trp53* locus in another sarcoma (2995) detected by deep sequencing was rarely found by Sanger sequencing of the cell lines derived from the same tumor (Fig. 3A, B). The low frequencies of two indels in *Trp53* found by Sanger sequencing suggests that it may have developed from independent clones (Fig. 3A). It is also possible that tetraploidization prior to mutagenesis may account for the detection of more than two distinct indels. In this scenario however, the frequency of indels generated in a tetraploid intermediate may be more evenly distributed than what we observed. Therefore, it is likely the sarcoma sample may have contained a dominant clone and other rare clones. These data also suggest that the endogenous genetic barcodes generated by Cas9 can be used to study tumor clonality at different stages of tumor development.

4. I am not convinced that the method presented is appropriate for barcode generation and tracking experiments. Given that the majority of CRISPR-induced mutations are 1bp deletions with high probability to occur at specific positions at the target site, it is impossible to predict relatedness of two cell clones based on the same 1bp mutation. This is only possible, if patterns of mutations are analyzed in a context of CRISPR-based multiplexed sgRNA delivery targeting multiple genes.

We agree that CRISPR-induced mutations are not random and are biased for individual sgRNAs, which was reported by several groups recently. We have carefully addressed this issue in the revised manuscript Discussion. We agree with Reviewer #2 that two clones sharing the same indels do not necessarily mean that these two clones are identical. In our experiments, we found that most of the mutations that we identified in each tumor were distinct from each other, suggesting that there is not one dominant mutation/indel class formed from this particular p53 sgRNA. Furthermore, we addressed this issue by using one sgRNA that targets two loci (Trp53 and chromosome 17) rather than using two or more sgRNAs that each target different loci. Thus, the indels present at the on-target chromosome 17 locus gives us another mutation site for our clonality study (i.e. 4 total alleles). Although this increases the power of our system, in the revised manuscript we do acknowledge that a multiplexed sgRNA approach targeting more than two sites in the genome would have greater sensitivity for true clonal populations.

Discussion, pages 13-15 (changes highlighted in yellow)

This system can also be used to study tumor clonality since unique indels are generated by NHEJ after gene editing by Cas9. It is important to note that variation between two indels may be unique in some cases, but are certainly non-random¹³. Consequently, targeting only one locus makes it impossible to assess clonality with any degree of certainty due to the probability of generating identical sets of indels in two or more independent clones due to indel bias. By targeting several loci and generating indels across multiple loci, a unique indel barcode can be generated endogenously. To demonstrate the feasibility of this application, we employed an sgRNA with two different perfect on-target sites, one in the Trp53 locus and another locus on chromosome 17, as a method of multiplexed generation of four unique indels per cell (one indel per targeted allele) that represents a specific and endogenous genetic barcode to follow clonality during soft tissue sarcoma development.

...

These data also suggest that the endogenous genetic barcodes generated by Cas9 can be used to study tumor clonality at different stages of tumor development. Here we demonstrate the feasibility of this approach using four genomic loci as potential endogenous barcodes. Multiplexed sgRNAs targeting more than four genomic loci will enable greater sensitivity for identifying and tracking truly clonal populations.

References

1. Platt, R. J. *et al.* CRISPR-Cas9 knockin mice for genome editing and cancer modeling. *Cell* **159**, 440–455 (2014).
2. Sánchez-Rivera, F. J. *et al.* Rapid modelling of cooperating genetic events in cancer through somatic genome editing. *Nature* **516**, 428–431 (2014).
3. Xue, W. *et al.* CRISPR-mediated direct mutation of cancer genes in the mouse liver. *Nature* **514**, 380–384 (2014).
4. Maddalo, D. *et al.* In vivo engineering of oncogenic chromosomal rearrangements with the CRISPR/Cas9 system. *Nature* **516**, 423–427 (2014).
5. Heckl, D. *et al.* Generation of mouse models of myeloid malignancy with combinatorial genetic lesions using CRISPR-Cas9 genome editing. *Nat. Biotechnol.* **32**, 941–946 (2014).
6. Weber, J. *et al.* CRISPR/Cas9 somatic multiplex-mutagenesis for high-throughput functional cancer genomics in mice. *Proc. Natl. Acad. Sci. U. S. A.* **112**, 13982–13987 (2015).
7. Chiou, S.-H. *et al.* Pancreatic cancer modeling using retrograde viral vector delivery and in vivo CRISPR/Cas9-mediated somatic genome editing. *Genes Dev.* **29**, 1576–1585 (2015).
8. Annunziato, S. *et al.* Modeling invasive lobular breast carcinoma by CRISPR/Cas9-mediated somatic genome editing of the mammary gland. *Genes Dev.* **30**, 1470–1480 (2016).
9. McFadden, D. G. *et al.* Mutational landscape of EGFR-, MYC-, and Kras-driven genetically engineered mouse models of lung adenocarcinoma. *Proceedings of the National Academy of Sciences* **113**, E6409–E6417 (2016).
10. Levine, D. S., Sanchez, C. A., Rabinovitch, P. S. & Reid, B. J. Formation of the tetraploid intermediate is associated with the development of cells with more than four centrioles in the elastase-simian virus 40 tumor antigen transgenic mouse model of pancreatic cancer. *Proc. Natl. Acad. Sci. U. S. A.* **88**, 6427–6431 (1991).
11. Margolis, R. L., Lohez, O. D. & Andreassen, P. R. G1 tetraploidy checkpoint and the suppression of tumorigenesis. *J. Cell. Biochem.* **88**, 673–683 (2003).
12. Fujiwara, T. *et al.* Cytokinesis failure generating tetraploids promotes tumorigenesis in p53-null cells. *Nature* **437**, 1043–1047 (2005).
13. van Overbeek, M. *et al.* DNA Repair Profiling Reveals Nonrandom Outcomes at Cas9-Mediated Breaks. *Mol. Cell* **0**, (2016).